# PARAMANU-GANITA: AN EFFICIENT PRE-TRAINED GENERATIVE MATHEMATICS LANGUAGE MODEL WITH CHAIN-OF-THOUGHT INSTRUCTION FINE-TUNING

## ABSTRACT

In this paper, we pose the following question: whether domain specific pretraining of tiny generative language models from scratch with domain specialized tokenizer and Chain-of-Thought (CoT) instruction fine-tuning results in very competitive performance on mathematical reasoning than LLMs which are trained on trillion of tokens and humongous parameters? Secondly, we pose our second RQ: whether domain specific pretraining from scratch is environmentally sustainable, highly cost efficient? To address these research questions, we present PARAMANU-GANITA, a 208 million-parameter novel Auto Regressive (AR) decoder based language model on *mathematics*. We performed pretraining from scratch on 31.5 billion tokens using a context size of 4096 on a mixed mathematical corpus consisting of mathematical web pages, mathematics related source code such as AlgebraStack, mathematical textbooks, Chain-of-Thought (CoT) templatised mathematical StackOverflow question answers pairs, and mathematical lecture notes in LaTeX curated by us. We also trained a math and code specialised BPE tokenizer. We proposed and performed Chain-of-Thought instruction fine-tuning of Paramanu-Ganita on the MetaMathQA dataset. We evaluate our model on GSM8K and MATH mathematical benchmarks, and on logical deductive reasoning (LogiQA) and multiple choice high school and college level math questions from SAT (AGIEVAL-SAT-Math), GRE/GMAT questions (AGIEVAL-AQuA-RAT), college and high school level math questions from MMLU. Our model Paramanu-Ganita, despite being 34 times smaller than the 7B LLMs, outperforms general LLMs by approximately 30% points, and even math-specialised LLMs by 3-23% points in GSM8K test accuracy metric. On MATH benchmark, Paramanu-Ganita outperformed the various models by 6-8% points. On other benchmarks such as LogiQA logical deductive reasoning benchmark, mathematical high school level multi-choice questions (MMLU-math-high-school), GRE-GMAT level quantitative questions (AGIEVAL-AQuA-RAT), SAT level math questions, Paramanu-Ganita was better than the others by about 1-4% points. The large significant margin improvement in performance of our math model over the existing LLMs signifies that reasoning capabilities of language models are just not restricted to those with humongous number of parameters. Paramanu-Ganita took only 170 hours of A100 training whereas large LLMs such as the math-specialised LLM, LLEMMA 7B, was trained for 23,000 A100 equivalent hours. Thus, our approach of pretraining powerful domain-specialised language models from scratch for domain adaptation is much more cost-effective and environmental friendly than performing continual training of LLMs.

## 1 INTRODUCTION

Pretrained Large Language Models (LLMs) such as LLaMa (Touvron et al., 2023a), LLaMa-2, (Touvron et al., 2023b), PaLM Chowdhery et al. (2023), Falcon (Almazrouei et al., 2023), Code LlaMa (Rozière et al., 2024), MPT (MosaicAI, 2023), etc. have demonstrated multi-dimensional abilities, such as in open-ended dialogue or instruction following capabilities (Ouyang et al., 2022).

Being typically generalist language models balancing the performance across the entire distribution of natural language tasks. However, these generalist models are humongous in size and requires millions of dollars to train aside from high engineering inference cost involved. Traditionally, to optimize performance within specific domains such as finance (Wu et al., 2023), medicine (Singhal et al., 2023), etc., these models have been continually trained on domain specific data. However, domain specific continual pretraining of LLMs are also very expensive as a lot of computation and inference costs are involved along with high requirement of GPUs. For example, to improve the mathematical reasoning capabilities of LLMs, LLEMMA 7B (Azerbayev et al., 2024) was trained on 256 A100 40GB GPUs for roughly 23,000 A100 training hours, which is extremely expensive.

In this paper, we search for a alternative approach to continual pretraining of LLMs for improving mathematical reasoning of LLMs like LLEMMA and cost-effective training and inference method. In particular, we try to answer the two following research questions.

**RQ1**: Is domain specific pretraining from scratch of tiny generative language model with domain specialised tokenizer and Chain-of-Thought (CoT) instruction fine-tuning results in competitive performance on mathematical reasoning than LLMs which are trained on trillion of tokens and humongous on the assumption that "Larger models trained on trillion tokens can only reason" parameters?

**RQ2:** Is domain specific pretraining from scratch of tiny generative language model is environmentally sustainable, highly cost efficient for both training and inference?

To answer these questions, instead of ~~Instead of~~ following the domain adaptation method of LLMs for better mathematical reasoning, we focused on *pretraining from scratch* a generative mathematical language model using only a high quality mathematical corpus curated by us. This avoids requiring immense compute power, high engineering maneuver and techniques to load LLMs in memory, and mostly high cost of training, and the misalignment of domain specialised tokenizers and embeddings with the existing embeddings of large language models (LLMs) via continual pretraining with vocabulary expansion of the existing LLMs tokenizers. We trained a powerful mathematical language model from scratch which required *only* 146 hours of A100 training and additional 14 hours for Chain-of-Thought (CoT) instruction fine-tuning. We call our model PARAMANU-GANITA[1]. Our model is based on the Transformer Decoder architecture (Vaswani et al., 2017). We have trained an auto-regressive model from scratch at a context size of 4096 on a single NVidia A100-PCIE-40GB GPU. Our models are small in size, having only 208 million parameters. Hence, our models are very fast in inference without requiring any quantization of weights, and our mathematics model inference can be run on CPU without need of GPU.

To test the mathematical problem solving ability of our tiny generative model, Paramanu-Ganita, we evaluated Paramanu-Ganita and compared with large generalist LLMs, code LLM, and math specialized LLMs across variety of grade level difficulty benchmarks including both discriminative multiple-choice math benchmarks across SAT, GRE, GMAT, graduate level, high school and grade level level math questions. We also tested our model on a logical reasoning benchmark (LogiQA).Table 2 and Table 3 shows the comparison of Paramanu-Ganita and LLMs ob various mathematical benchmarks. Although small, our mathematical language model, Paramanu-Ganita, still outperformed LLEMMA 7B math specialised model on GSM8K (Cobbe et al., 2021) benchmark by significant margin of 3 percentage points despite being 35 times smaller in size. On the memory requirements, the LLEMMA 7B checkpoint size is 13.5 GB whereas our model's checkpoint size is 2.5 GB and less than 1 GB in binary format (.bin). We found that our approach is highly cost efficient as we only spent on total 170 A100 hours including both pretraining from scratch and CoT fine-tuning, making our approach to be highly cost efficient, very competitive performance wrt LLMs, and least carbon footprint compared to LLMs or even math domain specialized LLM like LLEMMA which took 23,000 A100 hours for continual pretraining of Llama 2 and yet its performance (36.40%) is lower than our model, Paramanu-Ganita (39.4%) on GSM8K. Therefore, with our novel approach, we cut down the training cost by 135 times without compromising the performance of the model on mathematical benchmarks compared to both generalist and math specialized LLMs.

~~Our model is based on the Transformer Decoder architecture (Vaswani et al., 2017). We have trained an auto-regressive model from scratch at a context size of 4096 on a single NVidia A100-PCIE-40GB GPU. Our models are small in size, having only 208 million parameters. Hence,~~

---

[1]Paramanu means "atom" while Ganita is "mathematics"

~~our models are very fast in inference without requiring any quantization of weights, and our mathematics model inference can be run on CPU without need of GPU.~~

Our main contributions in this work are as follows:

1. We have curated a pretraining corpus for mathematics with high quality mathematical text from various public sources and in-house university lecture notes in LaTeX, textbooks, web crawled mathematical text, mathematical source code from various programming languages (AlgebraStack), and Chain-of-Thought (CoT) (Wei et al., 2023) templatised mathematical question answers pairs from forums like StackExchange.

2. We have developed a specialised tokenizer from scratch for mathematics domain and code.

3. We have developed an auto regressive decoder-only mathematical model, called Paramanu-Ganita, of 208 million parameters by pretraining from scratch on 31.5 billion tokens using a mixed corpus of mathematical text, source code, CoT templatised mathematical question answers at a context size of 4096 on a single Nvidia A100-PCIE-40GB GPU. We have also performed Chain-of-Thought (CoT) instruction fine-tuning of our model on MetaMathQA (Yu et al., 2024) dataset.

4. Our model, Paramanu-Ganita 208M, outperformed LLaMa-1 (33B, 13B, 7B), LLaMa-2 (7B, 13B), Falcon (40B, 7B), PaLM (62B, 8B), MPT (30B, 7B), Vicuna 13B, and math-specialised LLMs like Minerva 8B, LLEMMA-7B on GSM8K, MATH, AGIEVAL-AQuA-RAT benchmarks despite being smaller by multiple orders of magnitude in size.

## 2 RELATED WORK

Mathematical reasoning plays a crucial role in artificial intelligence, enabling the comprehension and resolution of intricate mathematical challenges. The incorporation of large language models (LLMs) in this area has been substantial, thanks to their capability to interpret, process, and produce complex natural language. In artificial intelligence, math problem solving involves utilizing algorithms, computational models, and use of increasingly LLMs to understand, explain, and resolve mathematical challenges. This method encompasses a wide range of topics, from basic arithmetic to advanced mathematics, including areas such as algebra, geometry, statistics, and calculus. (Wei et al., 2023) boosts the reasoning capacity of LLMs by supplementing the output with a series of intermediate steps leading to the answer. Several approaches have been suggested to enhance the quality of these reasoning paths. For instance, complexity-based CoT (Fu et al., 2023) picks examples with more steps as in-context demonstrations, demonstrating that prompting with additional reasoning steps improves performance. Self-consistency (Wang et al., 2023b) generates multiple reasoning paths and selects the final answer through majority voting. Another set of techniques involves fine-tuning-based methods, which adapt open-source models (like LLaMA) using insights from advanced closed-source LLMs (GPT-4, GPT-3.5-Turbo). (Magister et al., 2023) explore the transfer of reasoning abilities through knowledge distillation. (Yuan et al., 2024) advocate for the use of rejection sampling fine-tuning (RFT) to enhance mathematical reasoning performance. WizardMath (Xu et al., 2024) introduces a reinforced evol-instruct method for strengthening reasoning abilities through supervised fine-tuning and PPO training (Schulman et al., 2017). MAmmoTH (Yue et al., 2024) integrates CoT and Program-of-Thought (Chen et al., 2023) rationales to teach LLMs how to utilize external tools (such as a Python interpreter) for solving mathematical problems. (Wang et al., 2023a) propose a constraint alignment loss for fine-tuning LLMs to improve calibration. Going beyond the improvement of mathematical abilities through fine-tuning, LLEMMA (Azerbayev et al., 2024) introduces the Proof-Pile-2 dataset, which combines mathematical texts and code. By continuously pre-training with Code Llama, the model is equipped to utilize Python interpreters and formal theorem provers, showcasing remarkable performance on the MATH benchmark.

## 3 BACKGROUND

### 3.1 ROTARY POSITION EMBEDDING (ROPE)

Transformer-based models rely on positional embeddings to encode position and relative location information of words in a text. *Rotary Position Embedding (RoPE)* is a position encoding technique

proposed by (Black et al., 2022). Instead of adding positional embeddings or relative positional embeddings to token embeddings, RoPE rotates the token embedding by a fixed factor ($\theta$) in the higher-dimensional space to encode relative positional embeddings. In other words, RoPE encodes the absolute positions with a rotation matrix and meanwhile incorporates the explicit relative position dependency in self-attention formulation. The intuition behind RoPE is that we can represent the token embeddings as complex numbers and their positions as pure rotations that we apply to them. If we shift both the query and key by the same amount, changing absolute position but not relative position, this will lead both representations to be additionally rotated in the same manner. Thus, the angle between them will remain unchanged and, thus, the dot product will also remain unchanged. By exploiting the nature of rotations, the dot product used in self-attention will have the property for preserving relative positional information while discarding absolute position.

## 3.2 MODEL FLOPs UTILIZATION (MFU)

Model FLOPs Utilization (MFU) (Chowdhery et al., 2023) estimate is the ratio of the observed throughput (tokens-per-second) relative to the theoretical maximum throughput of a system at peak FLOPs. Model flops utilization (MFU) estimate the number of flops (floating point operations) done per iteration. It quantifies how efficiently the GPUs are utilized in model training.

## 3.3 MAXIMAL UPDATE PARAMETERIZATION

As the size of large language models (LLMs) and the scale of the dataset used in pretraining are expensively large, it is not feasible to perform hyperparameter tuning in LLMs. Yang et al. (2021) used a technique called maximal update parameterization ($\mu P$) to transfer the hyperparameters learnt from tuning of a small model to a larger model and found that the optimal hyperparameter values become stable across neural network sizes when the models have been parameterized using ($\mu P$).

# 4 DATA

We followed past works ((MA et al., 2024), (Razeghi et al., 2024), (Aryabumi et al., 2024)) that suggest that source code with text in the pretraining corpus improves the general and mathematical reasoning abilities of generative language models. Thus, we mixed source code related to mathematical problems along with open source mathematical web corpus and clubbed it with our curated lecture notes, and templatised mathematical questions answers in the pretraining dataset. Our pretraining dataset is a set of selected corpus from various publicly available datasets (AlgebraStack (Azerbayev et al., 2024), MathPile Commercial (Wang et al., 2023c), AutoMathText (Zhang et al., 2024), and Chain-of-Thought (CoT) templatised StackOverflow math, physics, statistics question answers (Zhang, 2024) and in-house collection of mathematical lecture notes in LaTeX. AutoMath-Text from AutoDS (Zhang et al., 2024) is a comprehensive and meticulously curated dataset that contains approximately 200 GB of mathematical texts. It is compiled from a variety of sources, including websites, arXiv, and GitHub (OpenWebMath, RedPajama, Algebraic Stack). This extensive dataset has been autonomously selected as zero-shot verifier and labeled by the advanced open-source language model, Qwen-72B. Each item in the dataset is assigned a score, lm_q1q2_score, ranging from [0, 1], which indicates its relevance, quality, and educational value in the field of mathematical intelligence.

For our pretraining data, we select only the web corpus from AutoMathText where the lm_q1q2_score $\geq$ 0.6. We selected textbooks, proofofwiki, wikipedia, and stackexchange subsets from MathPile Commercial dataset. We selected the source code from AlgebraStack. Finally, the pretraining corpus is composed of mathematical text from web, source code related to mathematical reasoning, one million question-answers pairs from StackOverflow, and in-house math lectures in LaTeX. Table 1 shows the number of words in our pretraining corpus. We used the following CoT template for templatising the StackOverflow and StackExchange questions answers as part of our pretraining corpus.

"Below is an instruction that describes a task. Write a response that appropriately completes the request. ### Q:{question} ### A: Let's think step by step. The answer is: {answer}"

| Pre-training Corpus | # Words |
|---|---|
| Web Corpus from AutoMathText | 1,245,273,066 |
| Math Pile Commercial | 854,692,279 |
| Math Code (AlgebraStack) | 678,729,775 |
| StackMathQA | 297,955,905 |
| Lecture Notes (ours) | 2,672,457,786 |
| **Total** | **5,749,108,811** |

Table 1: Pretraining Corpus

Following (Kocetkov et al., 2023), we removed duplicates and near-duplicates from the training data using (Mou et al., 2023), with default parameters. Following (Guo et al., 2024), we ran the data decontamination check in order to remove data contamination in the pretraining corpus from the various benchmark evaluations that we performed to test the performance of our language models. The filtering criterion is as follows: any text segment containing a 8-gram string that matches exactly with any sub-string from the evaluation benchmarks is removed from our pretraining corpus. For benchmark texts that are shorter than 8 grams but have at least 2 grams, we employ exact matching to filter out contaminated examples. This decontamination process leads us to remove around 170,346,325 words. Finally, the pretraining corpus has 5,578,762,486 words in the corpus.

## 5 TOKENIZATION

We trained two separate Byte-Pair encoding (BPE) (Sennrich et al., 2016) tokenizers using Sentencepiece (Kudo & Richardson, 2018) module on the pretraining data from scratch to develop mathematical domain specialised tokenizer to learn the intricacies of mathematical terminology.

One BPE tokenizer is trained on AlgebraStack (mathematical source code corpus) and another BPE tokenizer is trained on the mathematical text, lecture notes, and StackOverflow question answers. During pre-tokenization, NFC normalization was performed on the processed data, digits are split into individual tokens and fall back unknown UTF-8 characters to byte granularity for improving the arithmetic learning ability of the pretrained model. We treat our data as a sequence of bytes rather than Unicode characters, and we include each of the 256 bytes as tokens.

We then merged the both mathematical tokenizer and code tokenizer by intersection by removing the duplicate tokens to develop our final tokenizer specialised in mathematics and code, compact, optimized, and effective. Tokenizer has special tokens like "⟨Q:⟩", "⟨A:⟩", "⟨tex⟩", "⟨/tex⟩", "⟨python⟩", "⟨/python⟩", "⟨c⟩", "⟨/c⟩", "⟨matlab⟩", "⟨/matlab⟩" "⟨haskell ⟩", "⟨/haskell⟩". The size of the final tokenizer is 17,357.

## 6 MODEL ARCHITECTURE

The model architecture of Paramanu-Ganita is based on Transformer decoder-only architecture. It uses RMSNorm (Zhang & Sennrich, 2019) as pre-normalizaion layer and an approximate version of GeGLU (Shazeer, 2020) activation function for non-linearity by replacing the standard ReLU non-linearity activation function in the feed-forward dense layers. The model, Paramanu-Ganita, uses multi-head attention (MHA). The dimension is 1024 with 15 layers, n_k_v_heads=16, and 16 attention heads with feedforward layer hidden dimension of 2752. Following (Chowdhery et al., 2023), we remove all biases from dense layers to improve the training stability of Paramanu-Ganita.

## 7 TRAINING

### 7.1 MODEL TRAINING

We have pretrained our math model, Paramanu-Ganita, from scratch at a context size of 4096 on our curated corpus. However, we have excluded training of our math model on ArXiv math papers as we believe that to learn basic mathematical concepts, and acquire mathematical logical reasoning, ArXiv math papers are not required as they generally meant to serve beyond high school level

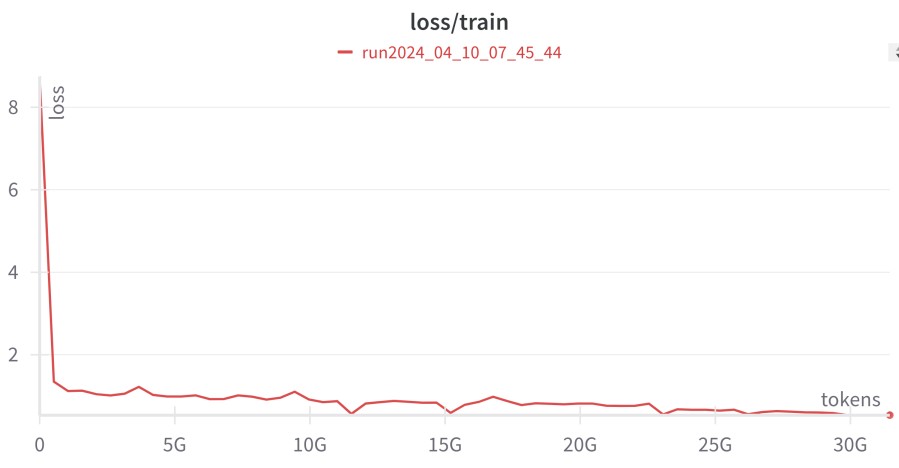

Figure 1: Training loss against number of tokens in billion. (G = billion)

mathematics. We started with simple strategy to use a part of our curated corpus which generally covers various mathematical and logical concepts till secondary school education in general. We performed mix pretraining combining both mathematical plain text, source code of programming languages, and templatised mathematical question answers pairs in the pretraining phase. For pre-training Paramanu-Ganita (4096 context size), we performed 95%-5% data split for pretraining. The perplexity of our model is 4.349 while the MFU is 40.392.

We performed hyperparameter tuning on 15M models to find the optimal vocabulary size, learning rate, learning rate scheduler, and warm-up ratio. We used a batch size of 8, gradient accumulation steps of 8, and the maximum sequence length set to 4096, i.e., 262,144 tokens per iteration. We used the concept of $\mu$ transfer, and transferred the learned hyperparameters to our bigger model for 208M Paramanu-Ganita from 15M model. Following (Hoffmann et al., 2022), we set $lr$ decay steps to $max\_steps$ and the minimum $lr$ is set nearly to $0.1 \cdot lr$. The $lr$ schedule starts with a linear warm-up from 0 to the maximum $lr$ at 1000 steps, followed by a cosine decay to the minimum $lr$ until the $max\_steps$ = 120,000 end of an epoch of training. We used the following equation for $lr$ decay ratio.

$$lr_{decay\_ratio} = \frac{t - warmup_{steps}}{lr_{decay\_steps} - warmup_{steps}}$$

where $t$ is the current training step. We set maximum learning rate ($lr$) to 3e-3 (max), weight decay to 1e-1. To further speedup training, we used BF16 mixed precision training. For our experiments and modeling, we implemented our code using Pytorch 2.0, in-house optmized CUDA kernels and used `torch.compile` feature for every model. We can see from Figure 1 how the loss is converging with incremental training steps and pretraining tokens, confirming a good pretraining with minor loss spikes. Paramanu-Ganita 208M is pretrained on around a total of 31.5 billion tokens. ~~Figure ?? shows the GPU power usage in Watt (W) and training hours during pretraining of Paramanu-Ganita. This illustrates the environment friendly nature of our model.~~

### 7.2 CHAIN-OF-THOUGHT INSTRUCTION FINE-TUNING

We performed Chain-of-Thought instruction fine-tuning on the MetaMathQA (Yu et al., 2024) instructions dataset, i.e, instead of regular instruction fine-tuning, we prepend the response of the MetaMathQA dataset by a prompt "Let's think step by step", and then used the prepended instruction, and response pair for instruction fine-tuning. We fine-tuned for 2 epochs due to limited computational resources. We used cosine learning rate scheduler with ($lr$) set to 2e-5 with gradient clipping of 1.0, warmup ratio of 0.05 and no weight decay. However, we believe our instruction-tuned Paramanu-Ganita would have performed better in benchmark evaluation if it was further fine-tuned for another 2-3 epochs. We used the following training prompt for MetaMathQA for our model.

"Below is an instruction that describes a task. Write a response that appropriately completes the request. ### Q:{query} ### A: Let's think step by step. {response}"

# 8 EVALUATION

In this section, we present results of our model Paramanu-Ganita against different LLMs, both general and code LLMs as well as math-specialized, on various benchmarks. We evaluated across variety of grade level of difficulty benchmarks including both discriminative multiple-choice math benchmarks across grade school level, high school level, college level, competitive exams level of SAT, GRE, GMAT, and math competition level questions. We also tested our model on a logical reasoning benchmark (LogiQA).

## 8.1 GSM8K AND MATH BENCHMARK DATASETS

We evaluate the model's ability to solve mathematics problems using chain of thought reasoning. Our evaluations include GSM8K (Cobbe et al., 2021) and MATH (Hendrycks et al., 2021b), which are the standard benchmarks for evaluating quantitative reasoning in language models. GSM8K includes 8,500 high-quality grade school math problems created by human writers. These problems generally consist of 2 to 8 steps to solve and mainly involve a series of basic arithmetic calculations to arrive at the final answer. The MATH dataset consists of 12,500 problems taken from high school math competitions. Each problem includes a step-by-step solution, allowing models to learn how to generate answer derivations and explanations. We used the following evaluation prompt for GSM8K test set for our math model.

"Below is an instruction that describes a task. Write a response that appropriately completes the request. ### Q:{question} ### A: Let's think step by step. The answer is: "

We used the vLLM (Kwon et al., 2023) inference engine and used the code from (Yu et al., 2024) for evaluation on GSM8K and MATH benchmarks. The following stop tokens were used while decoding from the model during evaluation. stop=['Question:', 'Question', 'USER:', 'USER', 'ASSISTANT:', 'ASSISTANT', 'Instruction:', 'Instruction', 'Response:', 'Response'] We set the vLLM inference engine parameters: best_of=8, presence_penalty=0.0, frequency_penalty=0.0, repetition_penalty=1.0, temperature=0.1, top_p=1, top_k=-1, min_p=0.0, length_penalty=1.0, max_tokens=1024 while decoding from Paramanu-Ganita for evaluation on GSM8K and Math test benchmarks. Answer extraction differs from the method used by (Wei et al., 2023) who rely on complex string rules to derive the final answer. In contrast, we follow the approach of WizardMath (Luo et al., 2023) by only extracting the string that follows "The answer is:" as the final answer. To train the model on this extraction technique, we append "The answer is: gold answer" to the end of the answers in the MetaMathQA dataset, replacing the gold answer with the corresponding answer for each question.

We report accuracy of Paramanu-Ganita and other models in Table 2. The scores of these models are reproduced as-is from their respective publications. Paramanu-Ganita, despite being 35 times smaller than the 7B family of LLMs, outperformed LLaMa-1 7B by 28.4% points, LLaMa-2 7B by 27.6% points, Falcon 7B by 32.6% points, PaLM 8B by 35.3% points, Minerva 8B by 23.2% points, and LLEMMA-7B by 3% points respectively. Paramanu-Ganita also outperformed PaLM 62B by 6.4% points despite being smaller by 305 times, Falcon 40B by 19.8% points (smaller by 192 times), LLaMa-1 33B by 3.8% points (smaller by 158 times), and Vicuna 13B by 11.8% (smaller by 64 times). This is a significant achievement since smaller models are preferred due to cost and environmental sustainability. Only the 3 giant LLMs, namely, LLEMMA 34B, Minerva 62B, Minerva 540B, performed better than Paramanu-Ganita on the GSM8K benchmark. On the MATH benchmark, Paramanu-Ganita outperformed LLaMa-1 7B by 7.44%, Llama-1 13B by 6.44% points, Llama-2 7B by 7.84% points, Llama-2 13B by 6.44% points, Falcon 7B by 8.04% points, Falcon 40B by 7.84% points, MPT 30B by 7.24% points, MPT 30B by 7.24% points, PaLM 8B by 8.84% points, and PaLM 62B by 5.94% points respectively. GPT-J and Vicuna did not report numbers for the MATH benchmark.

## 8.2 MULTIPLE-CHOICE MATH QA BENCHMARK DATASETS

We evaluate our model and compare with LLMs including general LLMs, math-specialized LLMs like LLEMMA, and code LLMs like CodeLlama on various multiple choice math question answers

| Model | Parameters | GSM8K | MATH |
|---|---|---|---|
| LLaMa-1 | 7B | 11.00 | 2.90 |
| LLaMa-1 | 13B | 17.80 | 3.90 |
| LLaMa-1 | 33B | 35.60 | 3.90 |
| LLaMa-2 | 7B | 14.60 | 2.50 |
| LLaMa-2 | 13B | 28.70 | 3.90 |
| Code Llama | 7B | 10.50 | 13.00 |
| Code Llama | 13B | 36.10 | 16.40 |
| Code Llama | 34B | 29.60 | 12.20 |
| Falcon | 40B | 19.60 | 2.50 |
| Falcon | 7B | 6.80 | 2.30 |
| MPT | 30B | 15.20 | 3.10 |
| MPT | 7B | 6.80 | 3.00 |
| GPT-J | 6B | 34.90 | - |
| Vicuna | 13B | 27.60 | - |
| PaLM | 8B | 4.10 | 1.50 |
| PaLM | 62B | 33.00 | 4.40 |
| Minerva | 8B | 16.20 | 14.10 |
| Minerva | 62B | 52.40 | 27.60 |
| Minerva | 540B | 58.80 | 33.60 |
| MAmooTH-CoT | 7B | 50.50 | 10.40 |
| WizardMath | 7B | 54.90 | 10.70 |
| MetaMath | 7B | 66.50 | 19.80 |
| LLEMMA | 7B | 36.40 | 18.00 |
| LLEMMA | 34B | 51.50 | 25.00 |
| Paramanu-Ganita | 208M | 39.40 | 10.34 |

Table 2: Evaluation of LLMs on GSM8K test set. PaLM (Chowdhery et al., 2023), LLaMa-1 (Touvron et al., 2023a), LLaMa-2 (Touvron et al., 2023b), Falcon (Almazrouei et al., 2023), Code LlaMa (Rozière et al., 2024), MPT (MosaicAI, 2023), Vicuna (Chiang et al., 2023), Minerva (Lewkowycz et al., 2022), MAmooTH-CoT (Yue et al., 2024), MetaMath (Yu et al., 2024), WizardMath (Luo et al., 2023), LLEMMA (Azerbayev et al., 2024) scores are quoted from respective author papers.

| Models | LogiQA | MMLU-math-high-school | MMLU-math-college | AGIEVAL-AQuA-RAT | AGIEVAL-SAT-Math |
|---|---|---|---|---|---|
| Llama-2 7B | 30.41 | 25.55 | 30.00 | 25.59 | 24.54 |
| CodeLlama-7B | 30.72 | 24.81 | 30.00 | 22.83 | 29.09 |
| OLMo 1B | 26.81 | 30.37 | 27.00 | 23.62 | 21.81 |
| LLEMMA 7B | 29.95 | 32.22 | 32.00 | 23.22 | 32.72 |
| Falcon 7B | 26.88 | 21.11 | 21.00 | 22.04 | 28.63 |
| Paramanu-Ganita 208M | 30.57 | 31.11 | 29.00 | 26.77 | 25.00 |

Table 3: Zero-shot evaluation of Paramanu-Ganita 208M and LLMs. All benchmark reports Accuracy except LogiQA, which reports Normalized Accuracy. We present the best score across our model checkpoints for Paramanu-Ganita. B=billion, M=million.

using lm-eval-harness (Sutawika et al., 2024) at zero-shot greedy decoding setting. We considered high school and college math MCQ question answers from MMLU (Hendrycks et al., 2021a), AGIEVAL-AQuA-RAT (GRE, GMAT 254 multiple-choice math questions taken from AQuA-RAT (Ling et al., 2017)) (Zhong et al., 2024), and AGIEVAL-SAT-Math (SAT 220 multiple-choice math questions). Table 3 compares Paramanu-Ganita with various LLMs. LogiQA (Liu et al., 2021) is a dataset created from various logical reasoning questions gathered from China's National Civil Servants Examination. Notably, LogiQA features bilingual questions in both English and Chinese, with the English version being a translation of the original Chinese text. We only considered the English version for evaluation.

## 9 RESULTS AND ANALYSIS

From Table 2 on GSM8K benchmark, Paramanu-Ganita, despite being 35 times smaller than the 7B family of LLMs, outperformed LLaMa-1 7B by 28.4% points, LLaMa-2 7B by 27.6% points, Falcon 7B by 32.6% points, PaLM 8B by 35.3% points, Minerva 8B by 23.2% points, and LLEMMA-7B by 3% points respectively. Paramanu-Ganita also outperformed PaLM 62B by 6.4% points despite being smaller by 305 times, Falcon 40B by 19.8% points (smaller by 192 times), LLaMa-1 33B by 3.8% points (smaller by 158 times), and Vicuna 13B by 11.8% (smaller by 64 times). This is a significant achievement since smaller models are preferred due to cost and environmental sustainability. Only the 3 giant LLMs, namely, LLEMMA 34B, Minerva 62B, Minerva 540B, performed better than Paramanu-Ganita on the GSM8K benchmark.

On the MATH benchmark as shown in the Table 2, Paramanu-Ganita outperformed LLaMa-1 7B by 7.44%, Llama-1 13B by 6.44% points, Llama-2 7B by 7.84% points, Llama-2 13B by 6.44% points, Falcon 7B by 8.04% points, Falcon 40B by 7.84% points, MPT 30B by 7.24% points, MPT 30B by 7.24% points, PaLM 8B by 8.84% points, and PaLM 62B by 5.94% points respectively. GPT-J and Vicuna did not report numbers for the MATH benchmark.

As shown in the Table 3 on LogiQA (Liu et al., 2021) benchmark, Paramanu-Ganita outperformed Llama-2 7B, OLMo 1B by 3.76% points, LLEMMA 7B, Falcon 7B by 3.69% points.

On mathematical high school questions (MMLU-math-high-school) benchmark as shown in the Table 3, Paramanu-Ganita outperformed Llama-2 7B by 5.56% points, CodeLlama 7B by 6.3% points, OLMo 1B, and Falcon 7B by 10% points but LLEMMA 7B outperformed Ganita by 1% point despite being 34 times larger in size.

On college level math questions (MMLU-math-college) benchmark as shown in the Table 3, Paramanu-Ganita 208M outperformed Falcon 7B by 8% points, OLMo 1B by 2% points, whereas both Llama-2 7B and CodeLlama 7B outperformed Paramanu-Ganita just by 1% point despite being 34 times larger.

On GRE-GMAT level quantitative questions (AGIEVAL-AQuA-RAT) benchmark as shown in Table 3, Paramanu-Ganita outperformed all the LLMs under comparison, i.e., Falcon 7B by 4.73% points, LLEMMA 7B by 3.55% points, OLMo 1B by 3.15% points, CodeLlama 7B by 3.94% points, and Llama-2 7B by 1.18% point respectively.

At SAT level math questions (AGIEVAL-SAR-Math) benchmark as listed in Table 3, Paramanu-Ganita outperformed Llama-2 7B, and OLMo 1B while lagging behind LLEMMA 7B by 7.72% points.

Despite being 35 times smaller in size, the performance of our model, Paramanu-Ganita, is comparable with the other LLMs. We believe the domain specific pretraining from scratch using high quality mathematical corpus of lecture notes, source code, web scrapped mathematical text and our Chain-of-Thought (CoT) templated formatted StackOverflow math, physics, statistics question answers along with our novel merged math and code specialized BPE tokenizer, and CoT instruction fine-tuning are the most probable causes to amplify the performance of strong mathematical reasoning in a tiny generative language model of 208 million parameters compared to LLMs which are

pretrained on all kinds of data whereas we focused only on mathematical and source code related to mathematics, mathematical question answers in COT template in our pretraining corpus. The major objective of this paper to see if domain adaptive pretraining from scratch is viable, highly cost efficient, sustainable alternative to continual pretraining of LLMs for domain generalization, in this paper, we performed end-to-end tiny/small 208 million parameters mathematical generative language model development from scratch and showed the advantage of our approach as supported by our empirical results.

## 10  CONCLUSIONS AND FUTURE WORK

In this paper, we presented an alternative approach to the hypothesis that LLMs can reason and they should be improved with continual pretraining and then fine-tuning approaches via reinforcement learning or vanilla supervised instruction-tuning for mathematical reasoning. Instead of continual pretraining of LLMs and then various fine-tuning approaches to improve the reasoning of LLMs, we introduce an exclusive tiny mathematical auto regressive decoder-based language model, Paramanu-Ganita 208M, which is pretrained from scratch only on mathematical mixed corpora of mathematical web text, textbooks, LaTeX lecture notes, math related programming source code and Chain-of-Thought (CoT) templatised mathematical question answers curated from various public sources for a context size of 4096. We proposed and performed CoT instruction fine-tuning of Paramanu-Ganita on MetaMathQA dataset. We evaluated our mathematical model on various grade level of difficulty multiple standard benchmarks across grade school, high school, college level, and competitive exams of SAT. GRE. GMAT and competition MATH benchmarks . We found that Paramanu-Ganita, despite being 35 times smaller than 7B LLMs, outperformed general LLMs and even LLMs built specially for mathematics, such as Minerva 8B and LLEMMA 7B, in accuracy metric in GSM8K and MATH benchmarks. We further evaluated our model on logical deductive reasoning LogiQA benchmark, SAT and GRE/GMAT mathematical multi-choice questions subset of AGIEVAL and on high school and college level math multi-choice questions from MMLU. We achieved comparable or better results than the competing 7B family of LLMs.

This exhaustive evaluation of our model and other LLMs on various level of difficulty of mathematical and logical deductive reasoning questions takes us to the conclusion that a tiny language model, when sufficiently trained enough from scratch with a domain specialised tokenizer, offers a more cost effective (using only 1 GPU, much less pretraining time, i.e., total 170 A100 hours) and environmental friendly (very less carbon footprint due to 170 A100 training equivalent) approach to even specialised domain expert models as it cut down the training cost by 135 times than the continual pretraining of Llama for mathematical reasoning ((Azerbayev et al., 2024)) without compromising the performance of tiny generative math model, Paramanu-Ganita, on mulitple math benchmarks. We are the first to show that such an approach works without limiting ourselves to the presumption that "bigger means stronger" and only working on top of LLMs without creating our own models from scratch.

For future work, we would like to extend our pretraining corpus with ArXiv math papers and perform additional reinforcement learning (RL) alignment such as DPO/PPO RL training with our mathematical model to see how the performance of our model improves after additional RL alignment.

## 11  ETHICS STATEMENT

We have used results of the other models from their respective publications. We have trained and evaluated our model on a single GPU. Thus, we do not see any issues of ethics for this work.

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
