# OpenReview forum: "Paramanu-Ganita: An Efficient Pre-trained Generative Mathematics Language Model with Chain-of-Thought Instruction Fine-Tuning"
_ICLR.cc/2025/Conference — Submitted to ICLR 2025_

### Official Review · Reviewer_MiTu · 2024-10-31

**Soundness:** 2
**Presentation:** 2
**Contribution:** 1
**Rating:** 3
**Confidence:** 5

**Summary:**

The author presents a small decoder-based language model on mathematics called Paramanu-ganita. They trained this model from scratch using the existing public mathematical corpus and also performed CoT instruction finetuning on top of it. They also train their own tokenizer specialised in math and code. Despite their model only having 208 million parameters, it outperforms general LLMs by approximately 30% points, and even math-specialised LLMs by 3-23% points in  GSM8K, 6-8% on MATH. The 208 million parameter model outperformed  LLaMa-1 (33B, 13B, 7B), LLaMa-2 (7B, 13B), Falcon (40B, 7B), PaLM (62B, 8B), MPT (30B, 7B), Vicuna 13B, and math-specialised LLMs like Minerva 8B, LLEMMA-7B on GSM8K, MATH, AGIEVAL-AQuA-RAT benchmarks. They also showed the reduced time and computation requirement to train this model as compared to existing LLMs.

**Strengths:**

1. A novel decoder model, that is 34 times smaller than existing LLMs and can outperform them by a huge margin
2. A detailed explanation of the training process required
3. Detailed benchmarking on GSM8K, MATH and other datasets.
4. Emphasis on the training time required and compared it to other existing LLMs, showing computation and environmental prowess in training an exclusive tiny model from scratch.

**Weaknesses:**

1. The paper uses Qwen-72B to label the corpus and use a score >= 0.6 for training the model, ensuring only a high-quality dataset is used. However, apart from this, the training process used is not novel. Specifically, there is no novelty in the model architecture or training paradigm used that can justify the complete novelty of the paper and also puts into question the improved performance of a 208 million parameter model over LLMs
2. The paper does not touch upon, newer and difficult mathematical datasets such as MATHBENCH or JEEBENCH. These are some datasets that were released after training cutoff time for some models, ensuring they are not part of their training data. These datasets are also much more difficult as compared to gms8k. This will ensure that the proposed model is robust in solving difficult problems that it hasn't seen before.
3. Will the checkpoint-filtered corpus used for training be publicly available?
4. How does the model perform on out-of-distribution data points, this can be checked by first doing a sanity check of data memorization/contamination [1]. Performing simple algorithms 1 and 2 from the paper will ensure that the model has not seen the evaluation dataset, making the results more robust.
5. The empirical analysis is missing from the paper. A thorough qualitative comparison of reasoning chains produced by Paramanu-Ganita versus other models on a few representative problems from the benchmark datasets. For example, what errors are made by existing LLMs vs. Paramanu-ganita and in which area does it improve?

Reference
[1] Golchin, Shahriar, and Mihai Surdeanu. "Time travel in llms: Tracing data contamination in large language models." arXiv preprint arXiv:2308.08493 (2023).

**Questions:**

Address the weaknesses of the paper mentioned above.

---

### Official Review · Reviewer_wvLy · 2024-11-03

**Soundness:** 1
**Presentation:** 1
**Contribution:** 1
**Rating:** 1
**Confidence:** 5

**Summary:**

The paper introduces PARAMANU-GANITA, a 208 million parameter mathematics-focused language model trained from scratch. The authors demonstrate that effective mathematical reasoning capabilities can be achieved with smaller, more efficient models when trained specifically for the domain. This approach offers significant advantages in terms of computational costs and environmental impact while maintaining good performance.

**Strengths:**

- Good empirical results despite smaller model size, demonstrating the effectiveness of their approach
- Demonstrates that smaller, more efficient models can achieve good mathematical reasoning performance

**Weaknesses:**

- The overall presentation of the paper still needs much improvement. The paper is not in ready-to-review or ready-to-submit status. The figures are pretty rough and unclear for what the authors want to express. For example, Figure 2 shows GPU Power Usage during pretraining of Paramanu-Ganita. But what conclusions do the authors want to make here? How does it illustrate the environment friendly nature of the model? For the figure 1, what does the blue line mean here?
- Limited Ablation Studies. The paper doesn't analyze the relative importance of different components of their training data (web text vs. code vs. lecture notes). It is unclear why the authors want to utilize these data sources and why the data mixture should be adopted as it is in the paper.
- Contamination issues. The model achieves good performance on GSM8K and MATH with 200M parameters. It is unclear whether there is data contamination issue.

**Questions:**

See weaknesses

---

### Official Review · Reviewer_ZyBe · 2024-11-04

**Soundness:** 3
**Presentation:** 2
**Contribution:** 2
**Rating:** 3
**Confidence:** 4

**Summary:**

Authors present a LLM specializing in mathematics, called Paramanu-Ganita. It is quite smaller in size and exhibits interesting performance benefits in several math and logical datasets, compared some LLMs with bigger size. Authors also trained tokenizers from scratch, curate a new dataset for pretraining and show that Paramanu-Ganita outperforms several general-purpose LLMs and some domain-specialist LLMs.

**Strengths:**

The paper is well-written. I appreciate the background and the description of how the model is trained. The idea of targeting mathematics is important and building LLMs specializing in math (at least some part of it) is important.

The dataset is an important contribution, however I am not sure whether the authors plan to make it public.

**Weaknesses:**

I feel the paper explores an interesting direction, but there are some concerns:

1. Firstly, GSM8k tests basic math word problem skills and given the model's GSM8k performance is pretty poor, I do not feel the model is ready yet. I think more experimentation is required. Also, how are Table 2 values computed? It seems the MetaMath paper reports GSM8K performance to be 82.3. Why is it 66.5 here? [1]

2. What is mostly missing from the paper are proper motivations and justification as to what "contributes" or what is expected to contribute to the "improved" performance?

 - Looking at this from a different point of view, why did the authors not start with MetaMath, then say change the tokenizers or change the dataset? Then, slowly demonstrate how all the innovations are truly necessary. At the least such ablations would have showed the necessity of new models.

 - Secondly, given the model's performance is not so great, what are we gaining by spending so much training time and cost?

3. One more important aspect is, what are the domains that the model targets? What are the grade levels? Is it the expectation that we will also do IMO problems starting from GSM8k? Or, are we targeting sub-disciplines algebra, pre-algebra, calculus etc.? I think this depth is also missing, so is related papers that investigate the need of such models [2].

[1] https://openreview.net/forum?id=N8N0hgNDRt
[2] MATHSENSEI: A Tool-Augmented Large Language Model for Mathematical Reasoning

**Questions:**

Some minor and major questions:
1. Abstract: Concrete examples would be better, such as which model did it beat despite being smaller etc..
2.  L196: Please give examples, what happens for various ways of writing floats. How are the European and US/UK numbers treated 1,43 vs 1.43.  Mixed numbers and digits, other mathematical symbols.  To me, its not so clear from the writing.
3. L210: The architecture description seems incomplete, given its a section. You have mentioned decoders elsewhere, but you should complete this here, mentioning how many layers of decoders (or ranges), how many dense layers, and some block diagrams, referred from the section. This is supposed to be the most important section.
4. L249: Whats the perplexity of other models, especially mathematics specialist or science specialist ones? Can you show a table comparing them? Otherwise, the standalone numbers do not make sense to me.

---

> ### Author Response · Authors · 2024-11-24
> **Rebuttal: Performance Analysis of Paramanu-Ganita v/s LLMs**
>
> Thank you for reading. We would like to point out that the performance of our 208M model is commendable given its size. As discussed in the paper, we are highlighting again the gain in performance of our approach while cutting down training cost by 135 times compared to LLEMMA 7B (ICLR'24) and the parameter count by 34 times.
>
> From Table 2 on GSM8K benchmark, Paramanu-Ganita, despite being 35 times smaller than the 7B family of LLMs, outperformed LLaMa-1 7B by 28.4\% points, LLaMa-2 7B by 27.6\% points, Falcon 7B by 32.6\% points, PaLM 8B by 35.3\% points, Minerva 8B by 23.2\% points, and LLEMMA-7B by 3\% points respectively. Paramanu-Ganita also outperformed PaLM 62B by 6.4\% points despite being smaller by 305 times, Falcon 40B by 19.8\% points (smaller by 192 times), LLaMa-1 33B by 3.8\% points (smaller by 158 times), and Vicuna 13B by 11.8\% (smaller by 64 times).
> This is a significant achievement since smaller models are preferred due to cost and environmental sustainability.
> Only the 3 giant LLMs, namely, LLEMMA 34B, Minerva 62B, Minerva 540B, performed better than Paramanu-Ganita on the GSM8K benchmark.
>
> On the MATH benchmark as shown in the Table 2, Paramanu-Ganita outperformed LLaMa-1 7B by 7.44\%, Llama-1 13B by 6.44\% points, Llama-2 7B by 7.84\% points, Llama-2 13B by 6.44\% points, Falcon 7B by 8.04\% points, Falcon 40B by 7.84\% points, MPT 30B by 7.24\% points, MPT 30B by 7.24\% points, PaLM 8B by 8.84\% points, and PaLM 62B by 5.94\% points respectively. GPT-J and Vicuna did not report numbers for the MATH benchmark.
>
> As shown in the Table 3 on LogiQA benchmark, Paramanu-Ganita outperformed Llama-2 7B, OLMo 1B by 3.76\% points, LLEMMA 7B, Falcon 7B by 3.69\% points.
>
> On mathematical high school questions (MMLU-math-high-school) benchmark as shown in the Table 3, Paramanu-Ganita outperformed Llama-2 7B by 5.56\% points, CodeLlama 7B by 6.3\% points, OLMo 1B, and Falcon 7B by 10\% points but LLEMMA 7B outperformed Ganita by 1\% point despite being 34 times larger in size.
>
> On college level math questions (MMLU-math-college) benchmark as shown in the Table 3, Paramanu-Ganita 208M outperformed Falcon 7B by 8\% points, OLMo 1B by 2\% points, whereas both Llama-2 7B and CodeLlama 7B outperformed Paramanu-Ganita just by 1\% point despite being 34 times larger.
>
> On GRE-GMAT level quantitative questions (AGIEVAL-AQuA-RAT) benchmark as shown in Table 3, Paramanu-Ganita outperformed all the LLMs under comparison, i.e., Falcon 7B by 4.73\% points, LLEMMA 7B by 3.55\% points, OLMo 1B by 3.15\% points, CodeLlama 7B by 3.94\% points, and Llama-2 7B by 1.18\% point respectively.
>
> At SAT level math questions (AGIEVAL-SAR-Math) benchmark as listed in Table 3, Paramanu-Ganita outperformed Llama-2 7B, and OLMo 1B while lagging behind LLEMMA 7B by 7.72\% points.
> Despite being 35 times smaller in size, the performance of our model, Paramanu-Ganita, is comparable with the other LLMs.
> We believe the domain specific pretraining from scratch using high quality mathematical corpus of lecture notes, source code, web scrapped mathematical text and our Chain-of-Thought (CoT) templated formatted StackOverflow math, physics, statistics question answers along with our novel merged math and code specialized BPE tokenizer, and CoT instruction fine-tuning are the most probable causes to amplify the performance of strong mathematical reasoning in a tiny generative language model of 208 million parameters compared to LLMs which are pretrained on all kinds of data whereas we focused only on mathematical and source code related to mathematics, mathematical question answers in COT template in our pretraining corpus.
>
> The major objective of this paper to see if domain adaptive pretraining from scratch is viable, highly cost efficient, sustainable alternative to continual pretraining of LLMs for domain generalization without dropping the performance, in this paper, we performed end-to-end tiny/small 208 million parameters mathematical generative language model development from scratch and showed the advantage of our approach as supported by our empirical results.

---

> > ### Author Response · Authors · 2024-11-25
> > **Rebuttal Revision**
> >
> > Thank you for reading our paper and rebuttal. We have incorporated your feedback and submitted a revision of the paper. We made changes in the Introduction and Abstract by introducing two RQs for stronger motivation as discussed before, adding motivation for our data selection, adding a section of Performance Analysis by replacing the analysis from Evaluation section, section of Data Contamination, and added 5 lines in the Conclusion section . We hope the presentation of the paper would be much better now as before. In the revision, the blue lines are the added lines whereas the removed lines are already crossed out.

---

### Meta-Review · Area_Chair_9uEQ · 2024-12-20

**Metareview:**

This paper presents Paramanu-Ganita, a 208M parameter math-focused language model trained from scratch, claiming competitive performance compared to much larger models while being more computationally efficient. While the empirical results show some promise, the paper has significant weaknesses: lack of thorough ablation studies to justify architectural choices, insufficient analysis of model performance and failure modes, missing qualitative comparisons of reasoning chains, and inadequate evaluation on newer challenging math benchmarks. The presentation is also messy and needs substantial improvement. Given these limitations and the reviewers' consensus on major concerns that were not adequately addressed in rebuttal, the paper is not ready for publication in its current form.

**Additional Comments On Reviewer Discussion:**

During rebuttal, reviewers raised concerns about: 1) Limited ablation studies and analysis to justify model design choices, 2) Messy presentation and unclear figures, 3) Need for evaluation on more challenging datasets, 4) Missing qualitative analysis of reasoning chains, and 5) Questions about training efficiency claims. While authors attempted to address these in rebuttal and revision by adding RQs, motivation, and contamination analysis, the core issues around missing ablations, empirical analysis, and clear presentation remain unaddressed. All reviewers maintained their negative assessment post-rebuttal, as the fundamental limitations were not resolved through the authors' responses or proposed revisions.

---

### Decision · Program_Chairs · 2025-01-22

Reject